# *Carpinus turczaninowii* Extract May Alleviate High Glucose-Induced Arterial Damage and Inflammation

**DOI:** 10.3390/antiox8060172

**Published:** 2019-06-11

**Authors:** Juhyun Song, So Ra Yoon, Youn Kyoung Son, Woo Young Bang, Chang-Hwan Bae, Joo-Hong Yeo, Hyun-Jin Kim, Oh Yoen Kim

**Affiliations:** 1Department of Anatomy, Chonnam National University Medical School, Gwangju 61469, Korea; juhyunsong@chonnam.ac.kr; 2Department of Food Science and Nutrition, Dong-A University, 37 550 beon-gil Nakdongdae-ro, Saha-gu, Busan 49315, Korea; yunsora0917@naver.com; 3Center for Silver-Targeted Biomaterials, Brain Busan 21 Plus Program, Dong-A University, 37 550beon-gil Nakdongdae-ro, Saha-gu, Busan 49315, Korea; 4Institute of Health Insurance and Clinical Research, National Health Insurance Service Ilsan Hospital, 100 Ilsan-ro, Ilsandong-gu, Goyang-si, Gyeonggi-do 10444, Korea; 5Biological Resources Utilization Department, National Institute of Biological Resources, 42 Hwangyeong-ro, Seo-gu Incheon 22689, Korea; sophy004@korea.kr (Y.K.S.); wybang@korea.kr (W.Y.B.); bae0072@korea.kr (C.-H.B.); y1208@korea.kr (J.-H.Y.); 6Department of Food Science & Technology, and Institute of Agriculture and Life Science, Gyeongsang National University, 501 Jinju-daero, Jinju 52828, Korea; hyunjkim@gnu.ac.kr

**Keywords:** *Carpinus turczaninowii*, human aortic vascular smooth muscle cells, high glucose, inflammation, arterial damage, phenolic compounds

## Abstract

Hyperglycemia-induced oxidative stress triggers severe vascular damage and induces an inflammatory vascular state, and is, therefore, one of the main causes of atherosclerosis. Recently, interest in the natural compound *Carpinus turczaninowii* has increased because of its reported antioxidant and anti-inflammatory properties. We investigated whether a *C. turczaninowii* extract was capable of attenuating high glucose-induced inflammation and arterial damage using human aortic vascular smooth muscle cells (hASMCs). mRNA expression levels of proinflammatory response [interleukin-6 (IL-6), tumor necrosis factor-α (TNF-α)], endoplasmic reticulum (ER) stress [CCAAT-enhancer-binding proteins (C/EBP) homologous protein (CHOP)], and adenosine monophosphate (AMP)-protein activated kinase α2 (AMPK α2)], and DNA damage [phosphorylated H2.AX (p-H2.AX)] were measured in hASMCs treated with the *C. turczaninowii* extracts (1 and 10 μg/mL) after being stimulated by high glucose (25 mM) or not. The *C. turczaninowii* extract attenuated the increased mRNA expression of IL-6, TNF-α, and CHOP in hASMCs under high glucose conditions. The expression levels of p-H2.AX and AMPK α2 induced by high glucose were also significantly decreased in response to treatment with the *C. turczaninowii* extract. In addition, 15 types of phenolic compounds including quercetin, myricitrin, and ellagic acid, which exhibit antioxidant and anti-inflammatory properties, were identified in the *C. turczaninowii* extract through ultra-performance liquid chromatography-quadrupole-time of flight (UPLC-Q-TOF) mass spectrometry. In conclusion, *C. turczaninowii* may alleviate high glucose-induced inflammation and arterial damage in hASMCs, and may have potential in the treatment of hyperglycemia-induced atherosclerosis.

## 1. Introduction

Diabetes mellitus (DM) has been known to cause cerebral vascular disorder and cardiovascular diseases (CVD), which are the major causes of death in patients with DM [1,2]. Several studies reported that the morbidity and mortality of DM patients are the result of vascular dysfunction [3,4,5]. DM patients had a four-fold higher risk for CVD compared to normal healthy subjects [6], and also showed elevated oxidative stress due to increased glucose oxidation, impaired redox cycle, enhanced glycation, and enhanced polyol pathway activity [7]. The increased oxidative stress in DM patients accelerates the progression of atherosclerosis and aggravates various cardiovascular events by promoting inflammation, blood plaque instability, endothelial dysfunction, and proliferation and migration of vascular smooth muscle cells (VSMCs) [8]. Although many factors contribute to the progression of atherosclerosis in DM patients, oxidative stress induced by hyperglycemia has been implicated as a potential mechanism that accelerates atherosclerosis progression [9,10,11]. In DM patients, excessive circulating glucose induces oxidative stress and promotes the production of the free radical-yielding superoxide anion (O_2_^−^) and hydrogen peroxide (H_2_O_2_) [12]. Hyperglycemia generates excessive reactive oxygen species (ROS), produced by the proton electromechanical gradient [9], resulting in severe diabetes-related vascular complications [3], DNA damage, and lipid peroxidation [13,14]. Several studies have shown that elevated glucose concentrations result in the impairment of an antioxidant defense in both diabetic animal models and DM patients [15]. The modulation of hyperglycemia may be a key to improve various diabetic pathologies and to prevent the progression of atherosclerosis by inhibiting vascular damage in DM patients.

Previous studies suggested that hyperglycemia in DM patients could be improved by natural antioxidant compounds such as reducers of glutathione, and vitamins E and C [16,17], which regulate oxidative stress and inflammatory processes [18,19]. Among the natural compounds, phytochemical compounds derived from plants have been considered important all over the world because of their roles in increasing antioxidant capacity, regulating enzyme activity, and modulating gene transcription [19,20]. *Carpinus turczaninowii* (*C. turczaninowii*), a member of the Betulaceae family, has been reported to play a role in antioxidant regulation and anti-inflammation modulation [21,22,23,24,25,26,27]. *Carpinus turczaninowii* (*C. turczaninowii*), a member of the Betulaceae family (Korean hornbeam) usually grown in the Jeju islands of Korea, has been reported to play a role in antioxidant regulation and anti-inflammation modulation [21,22,23,24,25,26,27]. Recent studies showed dose-dependent inhibition of nitric oxide (NO) and interleukin (IL)-6 productions from LPS-stimulated RAW264.7 cells (1 μg/mL of LPS) by the extract of *C. turczaninowii* branches and leaves (12.5 to 200 μg/mL) [21,22,23,24]. Antibacterial properties and skin-whitening (murine melanoma cells) were also observed in the *C. turczaninowii* leaf extract [24]. In our recent study, a complete extract of *C. turczaninowii* was found to contain high amounts of total phenolic compounds (225.6 ± 21.0 mg of gallic acid equivalents/g of the extract), as well as strongly scavenged free radicals (average 14.8 ± 1.97 μg/mL IC_50_ at 40 min) [27]. In addition, mRNA expressions of interleukin-6 (IL-6) and tumor necrosis factor (TNF)-α in human aortic vascular smooth muscle cells (hASMCs) were significantly suppressed by the extracts (1 and 10 μg/mL) at 6 h after exposure, and IL-6 secretion was dose-dependently suppressed at 2 h and 24 h after incubation with the extract at 1–10 μg/mL in non-stimulated and LPS-stimulated cells [27]. Given that *C. turczaninowii* has antioxidant capacity, we hypothesized that it may protect arterial cells from the hyperglycemia-induced oxidative stress, but the mechanism is not fully understood. Thus, we investigated how the *C. turczaninowii* extract modulates arterial inflammatory response and damage under high glucose conditions using human aortic VSMCs (hASMCs).

## 2. Materials and Methods

### 2.1. C. turczaninowii Extract Preparation

*C. turczaninowii* (branches, leaves, and trunk) were exclusively collected from Suin Mountain (GangjinGun, Korea) in January 2015, and identified by a plant taxonomist and curator associated with the Natural History Museum of Hannam University (Daejeon, Korea, specimen deposition #: NIBRVP0000519846). Briefly, the plant material was air-dried, ground, and extracted three times with 70% ethanol for 24 h at room temperature. The extract was filtered, evaporated under reduced pressure, freeze-dried to obtain a powder, and then stored in a deep freezer (−80 °C) before testing. For the experiments, the lyophilized *C. turczaninowii* extract powder was dissolved in 70% ethanol and filtered (0.2 μm Minisart^®^ syringe filter, Sartorius Stedim Biotech GmbH, Goettingen, Germany). The extract stock solution (final concentration: 30 mg/mL) was then aliquoted and stored at −80 °C for further analysis.

### 2.2. Cell Culture and Treatment Condition

Primary human aortic smooth muscle cells (hASMCs) (ATCC PCS-100-012, American Type Culture Collection, ATCC, Manassas, VA, USA) were maintained in a humidified atmosphere of 37 °C, with 5% CO_2_ in VSMC basal medium (without glucose and phenol red) (ATCC^®^ PCS-100-030™) supplemented with recombinant human basic fibroblast growth factor (5 ng/mL), rhInsulin (5 µg/mL), recombinant human epidermal growth factor (5 ng/mL), L-glutamine (10 mM), ascorbic acid (50 µg/mL), fetal bovine serum (5%), gentamicin (10 µg/mL), penicillin (10 Units/mL), streptomycin (10 µg/mL), amphotericin B (0.28 µg/mL), and phenol red (33 µM) (ATCC). To induce clinically hyperglycemic condition, we stimulated the cells with 25 mM (450 mg/dL) of glucose. Based on our previous study for cell viability [27], we used 1 and 10 μg/mL concentration of *C. turczaninowii* extract in this study.

### 2.3. Quantitative Real Time-PCR

To examine mRNA expression of interleukin (IL)-6, tumor necrosis factor-α (TNF-α), CCAAT-enhancer-binding proteins (C/EBP) homologous protein (CHOP), and adenosine monophosphate (AMP)-protein activated kinase α2 (AMPK α2) in hASMCs, we performed quantitative real time-polymerase chain reaction (qPCR). hASMCs were treated with the extract (final concentration: 1 and 10 μg/mL) under high glucose condition (25 mM) or not, and then incubated for 6 h. Briefly, total cellular RNA was extracted from hASMCs using TRIzol reagent (Invitrogen, Carlsbad, CA, USA) according to the manufacturer’s protocol. Poly (A) was added using poly (A) polymerase (Ambion, Austin, TX, USA). One Step SYBR^®^ Prime Script TM RT-PCR Kit II (Takara, Otsu, Shiga, Japan) was used to conduct the qPCR reaction. PCR was performed using the following primers (5′ to 3′): TNF-*α* (F): CGT CAG CCG ATT TGC TAT CT, (R): CGG ACT CCG CAA AGT CTA AG; IL-6 (F): GTT GCC TTC TTG GGA CTG AT, (R): CTG GCT TTG TCT TTC TTG TTA T, CHOP: (F): AGA ACC AGG AAA CGG AAA CAG A (R): TCT CCT TCA TGC GCT GCT TT, AMPK α2 (F): GCT GTG GAT CGC CAA ATT AT, (R): GCA TCA GCA GAG TGG CAA TA, and β-actin (F): TCT GGC ACC ACA CCT TCT A, (R): AGG CAT ACA GGG ACA GCA C. The PCR was performed at 42 °C for 5 min, 95 °C for 10 s, followed by 40 cycles of 95 °C for 15 s, 60 °C for 34 s, and 65 °C for 15 s. The mRNA expression of each transcript was assessed on an ABI prism 7500 Real-Time PCR System (Life Technologies Corporation, Carlsbad, CA, USA) and analyzed with comparative Ct quantification. β-actin was amplified as an internal control. The values were presented by relative quantity (RQ). Three wells were used in each condition per experiment, and all the experiments were repeated three times. 

### 2.4. Immunocytochemistry

hASMCs treated with the extract under high glucose condition or not were washed three times with PBS and permeabilized for 20 min. The cells were incubated with the primary antibodies for 16 h at 4 °C. Anti-rabbit phosphorylated H2.AX (1:500, Cell Signaling, Danvers, MA, USA) was used as the primary antibody. After a 16 h incubation, hASMCs were washed three times with PBS and incubated with each specific secondary antibody for 1.5 h at room temperature. The cells were washed three times for 3 min with PBS and were counterstained with 1 μg/mL 4′,6-diamidino-2-phenylindole (DAPI, 1:200, Invitrogen) for 10 min at room temperature. Cells were imaged using a Zeiss LSM 700 confocal microscope (Carl Zeiss, Thornwood, NY, USA).

### 2.5. UPLC-Q-TOF MS: Identification of Phenolic Compounds from C. turczaninowii Extract 

Phenolic compounds from *C. turczaninowii* were analyzed by an ultra-performance liquid chromatography-quadrupole-time of flight (UPLC-Q-TOF) mass spectrometry (MS) (Waters, Milford, MA, USA). The extract was injected into an Acquity UPLC BEH C18 column (2.1 mm × 100 mm, 1.7 um; Waters) at a column temperature of 40 °C. Mobile phase consisted of water with formic acid (FA, 0.1%) (A) and ACN with FA (0.1%, B) at a flow rate of 0.35 mL/min for 9 min. The eluents were ionized by electrospray ionization with negative mode and analyzed using a Q-TOF MS. The scan range of TOF MS data was from 50 to 1500 *m*/*z*, with a scan time of 0.2 s. The capillary voltage was set at 2.5 kV for negative mode, while the sample cone voltage was 40 V. The desolvation flow rate was 900 L/h at 400 °C and source temperature set to 100 °C. Leucine-enkephalin ([M − H] = *m*/*z* 554.2615) was used as a reference for lock mass at a frequency of 10 s. The MS/MS spectra were obtained using collision energy ramps from 20 to 45 eV. Metabolites were identified by the Unifi software (Waters, Milford, MA, USA) with various LC/MS databases and by published papers. 

### 2.6. Statistical Analysis

Statistical analyses were carried out using Win SPSS, ver22.0 (Statistical Package for the Social Science, SPSS Inc., Chicago, IL, USA). Data were expressed as the Mean ± SD of three independent experiments. For descriptive purposes, the mean values are presented using untransformed values. The differences between the groups were determined by an independent *t*-test. A two-tailed value of *p* < 0.05 was considered statistically significant.

## 3. Results

### 3.1. C. turczaninowii Extract Suppresses the mRNA Expression of Inflammatory Cytokines and the ER Stress Marker in hASMCs under High Glucose Conditions In Vitro 

To assess the mRNA expression of pro-inflammatory cytokines (TNF-α and IL-6) and an endoplasmic reticulum (ER) stress marker (CHOP) in hASMCs, we performed qPCR (Figure 1). Our results showed that the mRNA levels of TNF-α and IL-6 increased in 25 mM glucose-treated hASMCs relative to the non-treated cells (Figure 1a,b), but the increased mRNA expression levels of TNF-α and IL-6 was attenuated by treatment with at 1 and 10 μg/mL of the *C. turczaninowii* extract (Figure 1a,b). The mRNA level of CHOP, an ER stress marker was also increased in 25 mM glucose-treated hASMCs relative to the non-treated cells (Figure 1c); however, the increased CHOP expression was markedly reduced by treatment with 1 and 10 μg/mL of *C. turczaninowii* extract, and lowered almost to the levels observed in the non-treated cells (Figure 1c). Our results indicate that the mRNA expression of pro-inflammatory cytokines and the ER stress marker, CHOP, may be suppressed by *C. turczaninowii* extract treatment.

### 3.2. C. turczaninowii Extract Reduces the Phosphorylation of H2.AX in hASMCs under High Glucose Conditions In Vitro 

To determine whether high glucose-induced DNA damage in hASMCs is modulated by treatment with the *C. turczaninowii* extract, we performed immunocytochemistry to measure the expression of p-H2.AX as a marker of activated DNA damage in the cell [28]. Our results revealed that the immunoreactivity of p-H2.AX in hASMCs increased under high glucose conditions, while the increased immunoreactivity of p-H2.AX in 25 mM glucose-treated hASMCs was attenuated by *C. turczaninowii* extract treatment (Figure 2A,B). Our data shows that the *C. turczaninowii* extract may inhibit DNA damage in hASMCs under high glucose conditions.

### 3.3. C. turczaninowii Extract Reduces the Expression of AMPK α2 in hASMCs under High Glucose Conditions In Vitro

To assess the expression of AMPK α2 in hASMCs under high glucose conditions, we measured mRNA levels of AMPK α2 in hASMCs through qPCR analysis (Figure 3). We observed a reduction in AMPK α2 mRNA levels in hASMCs under high glucose conditions. However, the reduced mRNA expression of AMPK α2 was recovered with *C. turczaninowii* extract treatment (Figure 3). This data indicates that the *C. turczaninowii* extract may promote the activation of AMPK signaling, and subsequently protect hASMCs against high glucose condition.

### 3.4. UPLC-Q-TOF MS Analysis of C. turczaninowii Extract: Phenolic Components with Anti-Oxidant and Anti-Inflammation Capabilities

We performed UPLC-Q-TOF MS analysis to identify the specific components that contribute to the beneficial effect of the *C. turczaninowii* extract on hASMCs under high glucose conditions. We were able to identify 15 phenolic compounds from the *C. turczaninowii* extract (galloylquinic acid, 3,4-digalloylquinic acid, gallotannin, three unidentified ellagitannins, ellagic acid, myricitrin, quercetin 3-arabinoside, quercitrin, kaempferol 3-rhamnoside, quercetin 3-(2″-galloylrhamnoside), 2″-O-glloylvitexin, quercetin and quercetin-3,7-dirahamnoside) (Table 1, Figure 4). Considering that these compounds are anti-oxidants [22,23,29,30], we summarize that the *C. turczaninowii* extract has various anti-oxidants, and may act as a modulator of inflammation.

## 4. Discussion

Vascular complications such as atherosclerosis are commonly observed in DM patients, and these ultimately lead to an increase in morbidity and mortality [31]. VSMCs have been known to play a critical role in the progression of diabetic vascular complications in the arterial intima [31]. Increased proliferation and migration of VSMCs aggravates the progression of atherosclerosis by converting a fatty streak to an irreversible blood plaque [11]. Several studies have demonstrated that the vascular complications caused by hyperglycemia results from abnormal proliferation and migration of VSMCs [32,33,34] causing DM induced atherosclerosis [34].

Many studies have attempted to discover natural compounds to treat vascular complications because of the relatively low occurrence of the adverse effects of these compounds when compared with the artificial drugs [35,36,37]. In this study, we observed that hASMCs under high glucose conditions remained viable when treated with *C. turczaninowii* at low concentrations (1 and 10 μg/mL). Considering that hyperglycemia reduces cellular function and viability [38], the protective effect of *C. turczaninowii* is necessary to attenuate the cellular damage of diabetic VSMCs. Furthermore, we found that increased expression of p-H2.AX in hASMCs under high glucose conditions was dramatically attenuated by *C. turczaninowii* treatment. These data indicate that *C. turczaninowii* may be protective against the DNA damage observed during hyperglycemia. Increased p-H2.AX indicates an increase in DNA damage in VSMCs [39], and suggests the activation of various inflammatory signaling in responses to the DNA double-strand breaks [40]. Oxidative stress promotes apoptosis of VSMCs [41]; therefore, we hypothesized that *C. turczaninowii* at low concentrations may inhibit apoptosis of VSMCs by modulating p-H2.AX even under high glucose-induced oxidative stress. However, p-H2.AX staining was also found in cytosol, as well as in the nuclei under high glucose condition. According to the report by Jung et al. [42], γH2.AX could be induced in cytosol as well as nucleus upon DNA damage when TrkA was overexpressed. TrkA plays an important role in cell survival/differentiation, and apoptosis, and could exhibit pleiotropic effects in nonneuronal cells when its protein level is high, even in the absence of its ligand binding [42]. In addition, Okamura et al reported that histone H1.2 can be translocated to mitochodria and associated with Bak, leading to apoptotic cell death (bleomycin-treated human squamous carcinoma cells) [43]. Although the reason why γH2AX is induced in cytosol and how it leads to cell death is unclear, we assumed that γH2AX cytosolic accumulation may be involved in the choice of cell death together with its role in nucleic DNA damage. Future study is needed to identify the phosphorylated H2.AX observed in cytosol.

Moreover, this study shows that *C. turczaninowii* attenuates the increased expression of pro-inflammatory cytokines (TNF-α and IL-6) and ER stress response (CHOP) genes in hASMCs under high glucose condition. TNF-α, a major inflammatory cytokine, plays a critical role in vascular atherosclerosis [44,45], and is expressed by VSMCs in atheromatous plaques [46]. It can boost the pro-apoptotic effect by activating monocytes, endothelial cells, and macrophages, and can be considered as an inflammatory index that indicates accelerated atherosclerosis [47,48,49]. TNF-α also stimulates VSMC proliferation and migration in blood vessels [50]. In addition, IL-6 promotes the growth and migration of VSMCs via the induction of PDGF production [51,52,53], and is related to cytoskeletal reorganization [54]. Given our results, we hypothesize that *C. turczaninowii* may lessen the inflammatory responses, inhibit apoptosis of VSMCs, and reduce the migration and proliferation of VSMCs. 

ER stress is known to be enhanced in arterial SMCs in response to atherosclerosis risk factors [55,56]. Several studies have demonstrated the expression of Grp78 as an ER stress marker in atherosclerotic lesions [57], and the increased expression of ER chaperones was found in VSMCs in atherosclerotic plaques [55]. The proliferation of VMSCs plays a critical role in plaque progression [58]. Based on our results, *C. turczaninowii* may alleviate ER stress response and ultimately inhibit the migration of VSMCs which leads to the formation of plaques.

In addition, AMPK consisting of three α, β, and γ subunits allows cells to sense changes in their energy status [59,60]. The α subunit has catalytic kinase domain that transfers phosphate from ATP to the target protein [61]. Phosphorylation of Thr^172^ on the α subunit of AMPK ultimately activates AMPK [61,62]. Recent studies have demonstrated that AMPK-mediated cellular signaling plays a protective role in CVD [63,64]. The phosphorylation of AMPK results in increased glucose uptake and metabolism, and fatty acid oxidation [65]. The activation of AMPK has been reported to block the migration of VSMCs through the regulation of cytoskeletal stability [66]. Based on these evidences, the modulation of AMPK activation can be highlighted as a target for the treatment of obesity and T2DM [67]. Especially, AMPK α2 is an important modulator in the migration and proliferation of VSMCs [68]. Deletion of AMPK α2 triggers aberrant activation of nicotinamide adenine dinucleotide phosphate oxidase, leads to endothelial dysfunction [69], and results in the increased risk of atherosclerosis [64]. Some studies have reported that deficiency of AMPK α2 aggravates VSMC damage [68] and promotes migration of VSMCs [70,71]. Moreover, deletion of AMPK α2 activates the nuclear factor κB2 (NF-кB2)/p52 NF-kB signaling [68,72]. Considering our results, we assume that *C. turczaninowii* may inhibit VSMC damage via the activation of AMPK under hyperglycemic condition. However, we need further study for the measurement of p-AMPK using Western blotting to elucidate clear mechanism.

Previous studies have identified chemical compounds of *C. turczaninowii* branches, but which were limited to a few compounds such as naringenin and quercetin glycosides (flavonoids) together with carpinotriols A and B [22,23,25]. In this study, we analyzed components in *C. turczaninowii* using UPLC-Q-TOF MS. We identified 15 types of phenolic components from *C. turczaninowii* including quercetin, myricitrin, and ellagic acid in major proportions. Quercetin (3,5,7,30,40-tetrahydroxyflavone), which is a commonly distributed plant flavonoid, has been reported to display biological properties including anti-oxidant [73], anti-inflammation [74], and anti-viral properties [75] against CVD [76]. Moreover, quercetin has been known to alleviate cell apoptosis and reduce ER stress response in vascular endothelial cells under high glucose condition [29]. One study demonstrated that quercetin also suppresses atherosclerotic inflammation by modulating NF-kB signaling [77]. In addition, myricitrin, as a plant flavonoid, is distributed in the roots of *Myrica esculenta*, *Myrica cerifera*, *Nymphaea lotus*, and some other plants [78]. Myricitrin has been reported to possess anti-oxidative and anti-inflammatory properties, and also protects various cells against stress condition [79]. It has been known to block the expression of vascular adhesion molecule in TNF-α-stimulated VSMCs [80], alleviate apoptosis of endothelial cells, and prevent the onset of atherosclerosis [81]. Recent studies have demonstrated that myricitrin exerts anti-oxidant function in the cells under hyperglycemia induced oxidative stress condition [82]. Furthermore, ellagic acid (2,3,7,8-tetrahydroxybenzopyrano [5,4,3-cde] benzopyran-5-10-dione) as a phenolic natural compound is abundantly present in fruits and nuts such as blueberries and walnuts [83,84]. Previous studies have reported the protective effect of ellagic acid based on its superoxide and hydroxide scavenging [85,86], anti-oxidant, and anti-inflammatory properties [87,88,89]. As mentioned above, all the main components of *C. turczaninowii* that were analyzed have anti-oxidant and anti-inflammatory properties. We need further studies on animal and human models to elucidate the major component of *C. turczaninowii* among the identified ones, which contributes to arterial protection. 

## 5. Conclusions

Given the evidence presented above, *C. turczaninowii* may be an effective anti-oxidant and anti-inflammatory regulator to protect VSMCs against hyperglycemia-induced damage. In addition, we suggest that *C. turczaninowii* should be noted as a natural therapeutic compound to prevent and treat atherosclerosis in patients with hyperglycemia and/or diabetes.

## Figures and Tables

**Figure 1 antioxidants-08-00172-f001:**
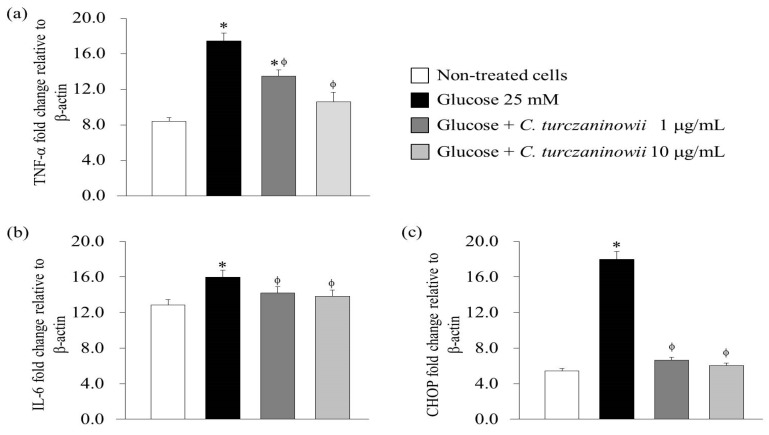
*C. turczaninowii* extract suppresses the mRNA expression of inflammatory cytokines and an endoplasmic reticulum (ER) stress marker in hASMCs under high glucose conditions in vitro. Data are expressed as mean ± SD. * *p* < 0.05 relative to with non-treated cells, and ^ϕ^
*p* < 0.05 compared with glucose-stimulated cells. The mRNA levels of TNF-α (**a**), IL-6 (**b**), and CHOP (**c**) were measured by reverse transcription PCR. Three wells were used in each condition per experiment, and all the experiments were repeated three times. *C. turczaninowii*: *Carpinus turczaninowii*; CHOP: CCAAT-enhancer-binding proteins homologous protein; hASMCs: human aortic vascular smooth muscle cells; IL-6: interleukin-6; TNF-α: tumor necrosis factor-α.

**Figure 2 antioxidants-08-00172-f002:**
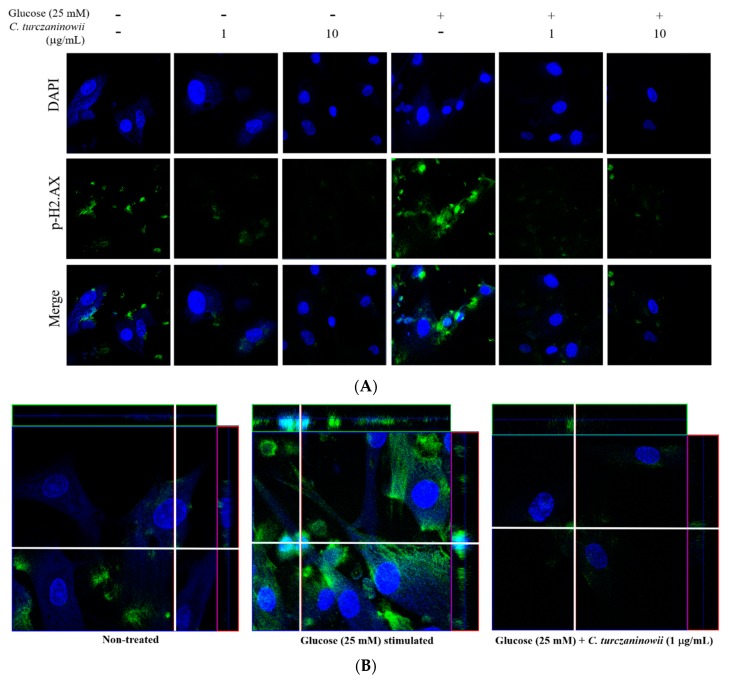
*C. turczaninowii* extract reduces the phosphorylation of H2.AX in hASMCs under high glucose conditions in vitro (**A**) with high magnification images (**B**). To confirm the effect of *C. turczaninowii* on the phosphorylation of H2.AX in high glucose-treated hASMCs, immunofluorescence staining was performed and analyzed by confocal microscopy. *C. turczaninowii*: *Carpinus turczaninowii*; hASMCs: human aortic vascular smooth muscle cells. Phosphorylated H2.AX is represented by green staining, nuclear DNA is indicated by 40,6-diamidino-2-phenylindole (DAPI) staining (blue color), and the combined images are presented.

**Figure 3 antioxidants-08-00172-f003:**
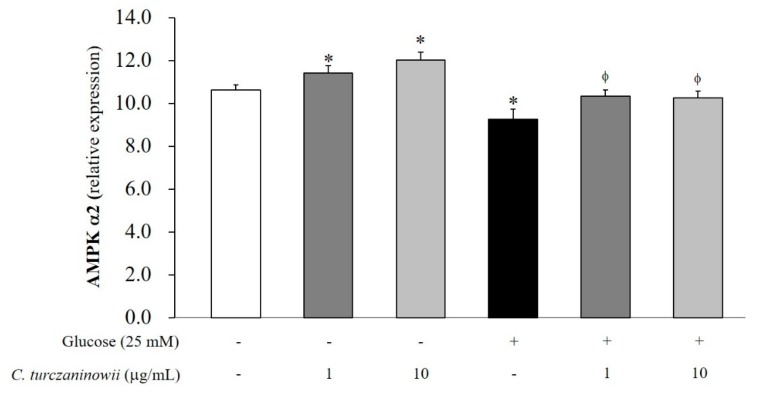
*C. turczaninowii* extract ameliorates AMP-protein activated kinase α2 (AMPK α2) of hASMCs under high glucose conditions in vitro. Data are expressed as mean ± SD. * *p* < 0.05 relative to non-treated cells, and ^ϕ^
*p* < 0.05 relative to glucose-stimulated cells. The mRNA levels of AMPK α2 were measured by reverse transcription PCR. Three wells were used in each condition per experiment, and all the experiments were repeated three times. *C. turczaninowii*: *Carpinus turczaninowii*; hASMCs: human aortic vascular smooth muscle cells; AMPK α2: adenosine monophosphate (AMP)-protein activated kinase α2; PCR: polymerase chain reaction.

**Figure 4 antioxidants-08-00172-f004:**
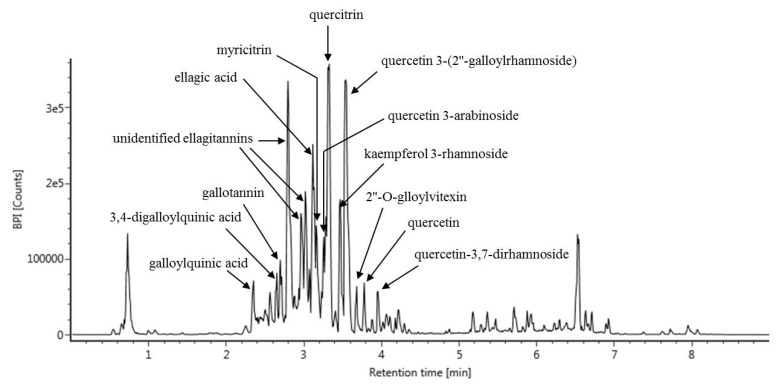
LC-MS chromatogram of *C. turczaninowii* extract and identification of major compounds. The extract was analyzed using a UPLC-Q-TOF MS with an Acquity UPLC BEH C_18_ column (2.1 mm × 100 mm, 1.7 μm) and negative electrospray ionization. The MS/MS spectra were obtained using collision energy ramps from 20 to 45 eV, and compounds were identified by the Unifi software with various LC/MS databases and by published papers. LC-MS: liquid chromatography mass spectrometry; MS/MS: tandem madss spectrometry.

**Table 1 antioxidants-08-00172-t001:** Identification of phenolic compounds from *C. turczaninowii* extract using ultra-performance liquid chromatography-quadrupole-time of flight mass spectrometry (UPLC-Q-TOF MS).

Retention Time (min)	Compound	*m*/*z* [M − H]	MS Fragments
2.35	Galloylquinic acid	343.0674	191, 169, 125
2.65	3,4-digalloylquinic acid	495.0782	343, 191, 169, 125
2.7	Gallotannin	633.0726	407, 301
2.79	Unidentified ellagitannin	951.0701	615, 301, 273, 245
2.96	Unidentified ellagitannin	953.0868	615, 301, 273, 246
3.02	Unidentified ellagitannin	965.0850	615, 301, 273, 247
3.12	Ellagic acid	300.9992	282, 271, 257
3.16	Myricitrin	463.0885	301, 300, 151
3.26	Quercetin 3-arabinoside	433.0779	301, 300, 271, 255
3.32	Quercitrin	447.0937	301, 300, 151
3.47	Kaempferol 3-rhamnoside	431.0985	327, 285, 284, 227, 255
3.54	Quercetin 3-(2″-galloylrhamnoside)	599.1037	429, 301, 178, 151
3.68	2″-O-glloylvitexin	583.1090	431, 345, 285, 125
3.78	Quercetin	301.0350	179, 151, 121
3.96	Quercetin-3,7-dirahamnoside	593.1293	447, 301, 300, 151

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
