# Peer review of "Carpinus turczaninowii Extract May Alleviate High Glucose-Induced Arterial Damage and Inflammation"

_antioxidants, 2019, doi:10.3390/antiox8060172_

Reviewer 1 Report

Authors have addressed the queries arose during the previous revision process.

I only have minor comments concerning this version of the manuscript:

- lines 222-224: This conclusion is not supported by the results, as cell  viability assay has been removed in this version of the manuscript.

- In the conclusion, add the sentence "Further studies on animal and human models are needed." or a similar sentence.

Author Response

Answers for Reviewer's comments.

The authors sincerely appreciate the time spent in reviewing this manuscript, and your advice to improve it. We revised the manuscript following your and reviewers’ queries and comments, and highlighted the corrected and revised parts of the text in red. Please, see the attached files (revised manuscript and point-by point answers to Reviewers’ comments). We hope that you find them satisfactory.

Reviewer (#1)' Comments to Author:

- lines 222-224: This conclusion is not supported by the results, as cell viability assay has been removed in this version of the manuscript.

Answer) Thank you for the reviewer’s comment. We removed the words “cell viability” from the sentence.

- In the conclusion, add the sentence "Further studies on animal and human models are needed." or a similar sentence.

Answer) In accordance with your advice, we revised the sentence to “We need further studies on animal and human models to elucidate the major component of C. turczaninowii among the identified ones, which contributes to arterial protection”.

Reviewer 2 Report

Dear Authors,

The manuscript falls with Journal s aim and scope. Authors have done a re-submission of a previous article submitted to other Journal. Compared to previous version the quality of manuscript has improved. However, Authors have to increase the resolution of Figure 2 and provide (as supplementary figure) the original immages for all presented data. Moreover, Author have to provide a rationale for use onòly 2 concentration of their extract next to the 3 concentratios/doses normally used in all biochemical/pharmacological research articles.

With Best Regards

Author Response

Answers for reviewers’ comments

Manuscript ID Antioxidants-510332R1

Carpinus turczaninowii Extract May Alleviate High Glucose Induced Arterial Damage and Inflammation

The authors sincerely appreciate the time spent in reviewing this manuscript, and your advice to improve it. We revised the manuscript following your and reviewers’ queries and comments, and highlighted the corrected and revised parts of the text in red. Please, see the attached files (revised manuscript and point-by point answers to Reviewers’ comments). We hope that you find them satisfactory.

Reviewer (#2)' Comments to Author:

The manuscript falls with Journal’s aim and scope. Authors have done a re-submission of a previous article submitted to other Journal. Compared to previous version, the quality of manuscript has improved. However, Authors have to increase the resolution of Figure 2 and provide (as supplementary figure) the original images for all presented data. Moreover, Author have to provide a rationale for use only 2 concentrations of their extract next to the 3 concentrations/doses normally used in all biochemical/pharmacological research articles.

Answer) The authors sincerely appreciate the reviewer’s comment for improving the manuscript. In accordance with your advice, we increased the resolution of Figure 2 and added more figures (Figure 2A and B) to help readers’ understanding. Regarding the concentrations applied in the experiments, As you pointed out, we used 1 and 10 mg/mL concentration of C. turczaninowii extract in this study based on our previous study for cell viability [ref-1 and 2]. Briefly, in our previous study, we tested cell viability with 5 dosages of the extract (1, 10, 30, 50, and 100 mg/mL). After 24 hrs, we found that the cell viabilities of human aortic vascular smooth muscle cells (hASMCs) incubated with 1 and 10 mg/mL of the extract were not significantly different from those of non-treated cells (CON) (about 104 % and 88 % respectively), but those incubated with 30, 50 and 100 mg/mL of the extracts were dose-dependently reduced compared with those of CON (about 80 %, 46 %, and 23 % respectively). Based on this result, we performed further molecular experiments with the extract whose concentrations between 1 and 10 mg/mL. This cell viability result in the previous study was poster-presented in the Planta Med (2016, ref-1), and will be published in Nutrition Research and Practice (2019 accepted, ref-2).

[ref-1] Yeo JH; Son YK; Bang WY; Kim OY. Carpinus truczaninowii extract showed anti-inflammatory response on human aortic vascular smooth muscle cells. Planta. Med. 2016, 82, S1-S381. (Poster presentation)

[ref-2] YK Son, SR Yoon, WY Bang, CH Bae, JH Yeo, R Yeo, J An, J Song, OY Kim. Carpinus turczaninowii extract modulates arterial inflammatory response: a potential therapeutic use for atherosclerosis. Nutr Res Pract, 2019 (accepted)

Reviewer 3 Report

Major

·      In the introduction, there is one sentence aboutCarpinus turczaninowii(line 68/69) with 7 references. Since this plant extract is the main topic of this publication, it’s background should be discussed in more detailPlease add a few sentences summarizing the results of the 7 studies that you cite. In addition, you refence your own work (Ref. 27, methods) stating that you already worked with this plant. Also include these results in the introduction. 

·      Results line 1263/Fig. 1/discussion/conclusion. The authors state that their extract reduced DNA damage under high glucose. However, it does so also under normal (? Info missing) glucose concentrations. These staining need to be quantified to support the author’s statements. It is certainly not enough to look at 1-2 cells per condition to draw firm conclusion.

·      I’m not familiar with h2.AX staining to assess DNA damage. But why is there excessive positive staining outside the nuclei and the nuclei themselves are negative? Isn’t the DNA in a cell in the nucleus (and little bit in the mitochondria)?

·      As its name suggests, p-AMPK gets phosphorylated. Total AMPK might be less important but the part that gets phosphorylated is the activate one. However, the authors did not assess phosphorylation (by Western blot) but they measured some kind of mRNA (AMPK?). It is not possible to asses p-AMPK by mRNA. Either the authors need to properly assess p-AMPK by WB or the need to correctly label what they have measured in the graph and in the text.

Minor

·      “glucose-induced damage” or similar wording. Sometimes it’s written with a hyphen, sometimes not (e.g. title). It should always be hyphenated.

·      What is the glucose concentration of the base medium? This info is missing in the methods section. Also, in Fig. 1, instead of “non-treated cells”, I would state “Glucose xx nM”

·      Fig 1. State how many times the experiment was done (it says repeated 3 times, but how many wells, etc.)

·      Line 224. “…and the extract showed no cellular toxicity”. The authors did not assess cytotoxicity.

·      Why are parts of the manuscript highlighted in yellow?

Author Response

Answers for reviewers’ comments

Manuscript ID Antioxidants-510332R1

Carpinus turczaninowii Extract May Alleviate High Glucose Induced Arterial Damage and Inflammation

The authors sincerely appreciate the time spent in reviewing this manuscript, and your advice to improve it. We revised the manuscript following your and reviewers’ queries and comments, and highlighted the corrected and revised parts of the text in red. Please, see the attached files (revised manuscript and point-by point answers to Reviewers’ comments). We hope that you find them satisfactory.

Reviewer (#3)' Comments to Author:

   · In the introduction, there is one sentence about Carpinus turczaninowii (line 68/69) with 7 references. Since this plant extract is the main topic of this publication, it’s background should be discussed in more detailPlease add a few sentences summarizing the results of the 7 studies that you cite. In addition, you refer your own work (Ref. 27, methods) stating that you already worked with this plant. Also include these results in the introduction. 

Answer) Thank you for the reviewer’s comment for improving our manuscript. In accordance with your advice, we briefly added some sentences explaining the results of the 7 studies and our work.

Carpinus turczaninowii (C. turczaninowii), a member of the Betulaceae family (Korean hornbeam) usually grown in the Jeju islands of Korea, has been reported to play a role in antioxidant regulation and anti-inflammation modulation [21-27]. Recent studies showed dose-dependent inhibition of nitric oxide (NO) and interleukin (IL)-6 productions from LPS-stimulated RAW264.7 cells (1 mg/mL of LPS) by the extract of C. turczaninowii branches and leaves (12.5 to 200mg/mL) [21-24]. Antibacterial properties and skin-whitening (murine melanoma cells) were also observed in the C. turczaninowii leaf extract [24]. In our recent study, complete extract of C. turczaninowii was found to contain high amounts of total phenolic compounds (225.6±21.0 mg of gallic acid equivalents/g of the extract), as well as strongly scavenged free radicals (average 14.8±1.97 μg/mL IC50 at 40 min) [27]. In addition, mRNA expressions of IL-6 and tumor necrosis factor (TNF)-a in human aortic vascular smooth muscle cells (hASMCs) were significantly suppressed by the extracts (1 and 10 mg/mL) at 6 hours after exposure, and IL-6 secretion was dose-dependently suppressed at 2 hrs and 24 hrs after incubation with the extract at 1 - 10 mg/mL in non-stimulated and LPS-stimulated cells [27].”

     · Results line 1263/Fig. 1/discussion/conclusion. The authors state that their extract reduced DNA damage under high glucose. However, it does so also under normal (? Info missing) glucose concentrations. These staining need to be quantified to support the author’s statements. It is certainly not enough to look at 1-2 cells per condition to draw firm conclusion.

Answer) The authors are sorry for making the reviewer confuse. We added Figure 2.B in the text to clarify the results and help the readers’ understanding. As shown in Figure 2.A below, we observed highly phosphorylated H2.AX as a marker of activated DNA damage in the cell in high glucose-stimulated cell. On the other hand, the treatment of C. turczaninowii extract significantly attenuated phosphorylation of H2.AX in the cell. Precisely, Figure 2.B shows that highly phosphorylated H2.AX was observed in the nuclei of the cells stimulated by high glucose condition, but the extract treatment significantly reduced phosphorylation of H2.AX. In addition, as you pointed out, significant reduction of phosphorylated H2.AX was also observed in non-stimulated cells (cotrol) after the extract treatment. In our previous study [27], we observed that pro-inflammatory cytokine (IL-6 and TNF-α) expressions in hAVSMCs without any stimulation (LPS or glucose stimulation) were reduced at 6 hrs after the extract treatment (1 and 10 μg/mL) compared with non-treated cells (control). Considering this result, we speculate that the antioxidant and anti-inflammatory effect of C. turczaninowii extract may reduce phosphorylation of H2.AX not only in the high glucose-stimulated cells, but also in the normal cells without any external stimulation.

· I’m not familiar with h2.AX staining to assess DNA damage. But why is there excessive positive staining outside the nuclei and the nuclei themselves are negative? Isn’t the DNA in a cell in the nucleus (and little bit in the mitochondria)?

Answer) Thank you for your comment. As mentioned in the above, phosphorylated H2.AX (gH2.AX) is well reported as an indicator of DNA damage in nucleus. However, as you commented, phosphorylated H2.AX staining was also found in cytosol as well as in the nuclei under high glucose condition (Figure 2B). According to the report by Jung et al. [ref-3], γH2.AX could be induced in cytosol as well as nucleus upon DNA damage when TrkA was overexpressed. TrkA plays an important role in cell survival, differentiation, and apoptosis, and could also exhibit pleiotropic effects in nonneuronal cells when its protein level is high, even in the absence of its ligand binding [ref-3]. In addition, Okamural et al reported that histone H1.2 can be translocated to mitochodria and associated with Bak, leading to apoptotic cell death, in the bleomycin-treated human squamous carcinoma cells [ref-4]. Although the reason why γH2AX is largely induced in cytosol and how it does lead to cell death is not clear at present, we assumed that γH2AX cytosolic accumulation may be involved in the choice of cell death together with its role in nucleic DNA damage. We also mentioned that future study is needed to identify the phosphorylated H2.AX observed in cytosol.

 [Ref-3] Jung EJ; Kim CW;Kim DR. Cytosolic accumulation of γH2AX is associated with tropomyosin-related kinase A-induced cell death in U2OS cell. Exp. Mol. Med. 2008, 40, 276-285.

[Ref-4] Okamura H; Yoshida K; Amorim BR; Haneji T. Histone H1.2 is translocated to mitochondria and associates with bak in bleomycin-induced apoptotic cells. J. Cell. Biochem. 2008, 103, 1488-1496.

 · As its name suggests, p-AMPK gets phosphorylated. Total AMPK might be less important but the part that gets phosphorylated is t measured he activate one. However, the authors did not assess phosphorylation (by Western blot) but they some kind of mRNA (AMPK?). It is not possible to asses p-AMPK by mRNA. Either the authors need to properly assess p-AMPK by WB or the need to correctly label what they have measured in the graph and in the text. 

 Answer) We appreciate your comment. As you pointed out, we measured mRNA expression of AMPK 2a but not do experiments to examine the phosphorylation of AMPK using Western blotting. We corrected it in Figure 3. Previous studies mentioned in the text reported that mRNA expression of AMPK 2a is related with the activation of AMPK, therefore, we thought that the mRNA expression of AMPK 2a indirectly explains the activation of AMPK. In the discussion, we mentioned the measurement of p-AMPK using Western blotting is needed for elucidating clear mechanism.

[Minor]

·  “glucose-induced damage” or similar wording. Sometimes it’s written with a hyphen, sometimes not (e.g. title). It should always be hyphenated.

Answer) In accordance with your advice, we hyphenated it (glucose-induced, glucose-treated, glucose-stimulated etc).

·  What is the glucose concentration of the base medium? This info is missing in the methods section. Also, in Fig. 1, instead of “non-treated cells”, I would state “Glucose xx nM”

Answer) For optimal growth of hAVSMCs (ATCC PCS­100­012, ATCC, Manassas, VA, USA), we used the VSMC basal medium and supplementation recommended by the manufacturer ATCC. As we mentioned in the method section, we used VSMC basal medium (ATCC® PCS­100­030™, ATCC) supplemented with VSMC growth kit which was composed of rhFGF-basic (5 ng/mL), rhInsulin (5 µg/mL), rhEGF (5 ng/mL), L-glutamine (10 mM), ascorbic acid (50 µg/mL), fetal bovine serum (5 %), gentamicin (10 µg/mL), penicillin (10 Units/mL), streptomycin (10 µg/mL), amphotericin B (0.28 µg/mL), and phenol red (33 µM) (ATCC PCS­100­042, ATCC). According to the product sheet, VSMC basal medium contains essential and non­essential amino acids, vitamins, other organic compounds, trace minerals and inorganic salts. It does not contain glucose. To clarify the compoents in VSMC basal medium, we inserted the words “(without glucose and phenol red)” right after ‘VSMC basal medium’.

·  Fig 1. State how many times the experiment was done (it says repeated 3 times, but how many wells, etc.)

Answer) In accordance with your comment, we clarified how many wells and how many time the experiment were done. We used 3 wells in each condition per experiment, and repeated the experiments 3 times.

“Three wells were used in each condition per experiment, and all experiments were repeated three times.

·  Line 224. “…and the extract showed no cellular toxicity”. The authors did not assess cytotoxicity.

Answer) The authors are sorry again for making the reviewer confused. We removed it.

·  Why are parts of the manuscript highlighted in yellow?

Answer) We removed them.

Round  2

Reviewer 3 Report

The authors have answered my questions and revised their manuscript accordingly.